# Artificial Intelligence in mental health and the biases of language based models

**Isabel Straw[1]\*, Chris Callison-Burch[2]**

**1** Department of Public Health, Perelman School of Medicine, University of Pennsylvania, Philadelphia, Pennsylvania, United States of America, **2** Computer and Information Science Department, University of Pennsylvania, Philadelphia, Pennsylvania, United States of America

\* isabelstraw@doctors.org.uk

**Data Availability Statement:** All relevant data are within the paper and its Supporting information files.

**Funding:** The author(s) received no specific funding for this work.

## Abstract

### Background

The rapid integration of Artificial Intelligence (AI) into the healthcare field has occurred with little communication between computer scientists and doctors. The impact of AI on health outcomes and inequalities calls for health professionals and data scientists to make a collaborative effort to ensure historic health disparities are not encoded into the future. We present a study that evaluates bias in existing Natural Language Processing (NLP) models used in psychiatry and discuss how these biases may widen health inequalities. Our approach systematically evaluates each stage of model development to explore how biases arise from a clinical, data science and linguistic perspective.

### Design/Methods

A literature review of the uses of NLP in mental health was carried out across multiple disciplinary databases with defined Mesh terms and keywords. Our primary analysis evaluated biases within 'GloVe' and 'Word2Vec' word embeddings. Euclidean distances were measured to assess relationships between psychiatric terms and demographic labels, and vector similarity functions were used to solve analogy questions relating to mental health.

### Results

Our primary analysis of mental health terminology in GloVe and Word2Vec embeddings demonstrated significant biases with respect to religion, race, gender, nationality, sexuality and age. Our literature review returned 52 papers, of which none addressed all the areas of possible bias that we identify in model development. In addition, only one article existed on more than one research database, demonstrating the isolation of research within disciplinary silos and inhibiting cross-disciplinary collaboration or communication.

### Conclusion

Our findings are relevant to professionals who wish to minimize the health inequalities that may arise as a result of AI and data-driven algorithms. We offer primary research identifying

**Competing interests:** Chris Callison-Burch declares no competing interests. Isabel Straw has been an unpaid intern with the e-Health company "Neuroflow" as part of her Masters in Public Health (MPH) Program. While this has been entirely separate from this research project, the timing of the internship overlaps with the period of time taken to write this article. This does not alter our adherence to PLOS ONE policies on sharing data and materials.

biases within these technologies and provide recommendations for avoiding these harms in the future.

## 1.0 Introduction

Our mental and emotional wellbeing is predominantly communicated through our language, and as a result psychiatric professionals have historically relied on clinical dialogue and patient narrative for assessing mental health. However, recent developments in Artificial Intelligence (AI) have brought new insights into the field through technologies that can infer emotional meaning from a more diverse range of data sources [1–3].

The disciplines of computational linguistics and sentiment analysis have been central to this process [2–5]. In computational linguistics 'Natural Language Processing' (NLP) is a technique used to build computational models that can interpret raw human language data [2–5]. Sentiment analysis is a subset of AI that measures, understands and responds to linguistic representations of human emotions. The combination of NLP and sentiment analysis has empowered data scientists to build models that can understand human emotion from written text [3]. For medicine, these models are now being used to provide rich information on the emotional and psychological wellbeing of patients [6–11].

Over the past few years NLP models have been used to identify suicidal ideation from clinical notes, predict suicide risk online and have mined for psychiatric self-disclosures on twitter [7,10–13]. These models have applications for both individual patient care and wider public health policy. Population level applications include NLP algorithms that effectively map behavioral health illnesses across the United States, correlating with public health data from the Center for Disease Control and Prevention (CDC) [8]. On an individual level, researchers have demonstrated high accuracy in predicting which mothers will suffer from postpartum depression using their online data [9,14].

Despite the enthusiasm surrounding these technologies they must be considered in the wider context of our existing healthcare transformation. The integration of digital health into medicine is bringing rapid changes to the healthcare field and the decisions we make now will have far reaching effects into the future of patient care. At present, researchers and developers are building these tools with the assumption that existing medical practice is 'gold standard', despite the field's long history of discriminatory practice, biases and medical error [15–23]. For example, the forty year Tuskegee trials demonstrated the history of racist research in medicine; the 'hysteria' diagnoses of the 20th century actively harmed women with organic disease; and the pathologisation of homosexuality reflects the longstanding sexual discrimination of the medical discipline [15,17,24]. The history of medicine is littered with examples of prejudicial harmful practice based on identity and we continue to see the impact of this history in clinical practice today. If we are to create models that do not harm disadvantaged patient groups we must first question the foundation on which these models are built.

The ongoing research of existing biases in medicine provides the ideal resource for doing this. Current public health research has illustrated that when women and men present with the same medical symptoms, women's symptoms are more likely to be interpreted as psychosocial leading to delayed treatment [15]. Pharmacological research demonstrates that the exclusion of minority populations from drug trials has led to interventions that do not benefit all patients at the same rate [15,25–27]. In addition, gender-based misdiagnoses due to biases in medical curricula and diagnostic frameworks have placed women at a greater risk of adverse outcomes from cardiac events [15,16].

In the field of psychiatry we see further issues of discriminatory practice. For patients with the same set of trauma symptoms women are more likely to receive personality disorder diagnoses, while men receive PTSD diagnoses [18–20]. Patients from racial minority backgrounds receive disproportionately high doses of psychiatric medications and are less likely to receive guideline adhering treatment [28]. While investigations into medical bias has grown substationally over the past ten years, researchers are stll only beginning to reveal the biases that exist within the mental health field.

Throughout this paper will build on existing medical bias research by evaluating biases that exist in digital mental health models, specifically those that use NLP to assess patient mental health. As discussed above, NLP is one of the predominant methods being employed by data scientists to develop algorithms that can assess mental health from online data. Throughout this paper we will examine components of these digital medical models and we will conclude by discussing how digital health professionals can prevent the exacerbation and projection of medical biases into the future of digital healthcare.

## 2.0 Background and literature review

Table 2.0 demonstrates three key stages in NLP model development where bias may materialize. The first stage is the data that is collected by researchers, which is typically drawn from large text databases. Social scientists and psychology researchers use social media sites such as Reddit, Facebook and Twitter. These data resources rely on the data that individuals express online and hence we have named the first stae of model development "Expression of Data" [2–3]. The second stage considers the AI model itself by exploring the constituents of the NLP algorithm. We term the second stage the "Analysis of Data" as this refers to the data analytic models that are used in the process. The third stage considers the interpretation of the model results by the practitioner and how this interpretation may be influenced by different factors. This third and final stage is named "Interpretation of results" (see Table 2.0). Presently, no paper has taken a multidisciplinary approach to comprehensively investigate each step for possible bias in NLP model development.

We undertook a systematic literature review of uses of NLP in mental health across multiple disciplinary databases to establish the present understanding of NLP bias from a linguistic, data science and clinical perspective. S1 Appendix details the MeSH terms and databases used to search for articles and S2 Appendix provides a full list of these referenced articles. In our method we extracted data from multiple different disciplinary databases as NLP research is undertaken in different disciplinary silos and there is little crossover. We used the following databases: ACL Anthology, PubMed, ArXiv, Scopus, Engineering Village and Association for Computing Machinery (ACM). Each article was assessed to see whether the authors had considered bias in each area of NLP model development for each of the three stages detailed in Table 2.0. In the following sections we provide a background on the importance of each stage of model development and the findings of our literature review concerning each area.

**Table 2.0. The stages in the development of a Natural Language Processing (NLP) model that are vulnerable to bias when used in the context of mental or emotional health.**

| 1. Expression of Data | 2. Analysis of Data | 3. Interpretation of Data |
|---|---|---|
| The language an individual uses to express their emotions online and the data that they produce. | The NLP algorithm itself including the chosen model, training data, features and word embeddings. | The existing injustices of established medical dogma and the implicit biases of interpreting practitioners. |

## 2.1 Expression of data

**Background.**   AI models that use NLP rely on large datasets of expressive language which are typically obtained from social media sites [2,3,6–12]. Researchers also obtain language data from online forums, blogs and chat rooms and use it to profile mental health [3]. Yet before collecting this data from the online community, it is essential that developers consider how this data is already influenced by an individual's personal background.

Language is an expression of our identity affected by our social context. Pennebaker et al. describe how language changes over the lifespan, varies by gender and is influenced by different personality traits [29]. In the psychiatric literature there exists extensive research that describes how depression closely relates to increased use of first person singular pronouns, however we are yet to see a tool that can accurately differentiate between word particles that result from mental pathology, and those that originate from individual personality differences [29,30].

Language is also shaped by culture. Desai and Chaturvedi describe how "idioms of distress" differ between cultural communities and detail how the psychological symptoms in traditional cultures may not fit within existing western psychiatric frameworks [31]. Non-western expressions of mental illness are often not captured in the mental health literature and will therefore be missed by NLP models that are based on our existing medical assessments [31–36].

Furthermore, gender plays a pivotal role. Men and women write suicide notes differently [37]. Lester & Lin describe the observable differences between the genders in expressing suicidal distress, both in lexical content and text theme [37]. Chaplin compares the heightened expression of positive emotions and internalizing of negative emotions by women, and the increased expression of anger by men [38]. An NLP model that screens for psychopathology or suicide for one gender, may be inappropriate for another (and this is considering gender in a binary context which excludes a large part of the population). If models are trained predominantly on data written by white men this will not accurately represent people in other demographic groups.

The influence of all the above factors come together to shape the language that we express. A simplistic NLP model that treats language expression as a homogenous dataset will be inaccurate for large portions of the population. Any model which aims to infer pathology from language, must begin by considering how that language is a result of its personal and social context, as opposed to assuming pathology.

**2.1.1 Literature review findings.**   Of the 52 articles that were examined only eight discussed the implications of how the language we express is influenced by our identity and demographic features. The articles that did discuss the influence of social factors predominantly focused on the influence of personality and educational level in on the outputs of the NLP models [9,10,39–45]. Of the eight studies, two studies provided a table of the demographic features of the initial dataset, however no studies stratified the outputs of their NLP models by by demographic features.

## 2.2 Analysis of data

In Section 2.1 we examined how the data fed into an NLP model may already be influenced by social factors, now we will focus on the means by which the model itself can introduce bias. To do this we must consider thie history of NLP model development and how modern NLP models differ from those used in the past. In the past computational linguistic models used 'lexical' approaches that sought to understand language by considering words in isolation [5,46]. Lexical approaches have been the mainstay of NLP models which use tokenization of text corpora and then attempt to infer the sentimental meaning of the corpus [5,46]. In recent years, more novel AI approaches have evolved which utilize deep learning and neural nets to understand

language and infer sentiment/meaning [32,41–45]. Part of this approach includes the use of 'word embeddings', which will be a key focus of this study.

Word embeddings provide a mathematical model for inferring semantic meaning from words [46–50]. The word embeddings are a key component of the NLP model and they rely on the distributional hypothesis which states "a word is characterized by the company it keeps". NLP models based on this concept use the relationships and distances between words to the infer meaning contained in a selection of text.

A word embedding consists of thousands of words, in which each word is represented by a geometric vector in a graphical space [48–50]. The distance between two words can be measured as a vector distance. This distance is then used to infer the relationship between two words. For example, in a word embedding space, America and New York are likely be closer to one another than America and Bangkok [48–50]. Words in similar contexts are found in similar regions of the vector space and are therefore expected to have similar meanings.

Existing research has illustrated biases that exist within the word embedding component of NLP models. These biases have predominantly been explored in the context of the social sciences [51]. Garg et al. provide a comprehensive analysis of word embedding models demonstrating historical trends in gender and ethnic stereotypes [51]. Furthermore, research from Bolukbasi et al. illustrate female/male gender stereotypes relating to occupation in their seminal study: "*Man is to Computer Programmer as Woman is to Homemaker*? *Debiasing Word Embeddings*" [49]. Yet, despite the exploration of word embedding bias in sociology, no paper has yet examined such biases in word embeddings that relate to mental health. To address this gap in the research, we follow up on our literature review with our own primary analysis in Section 3.0 of this paper. Section 3.0 evaluates biases in two commonly used word embeddings–'GloVe' and 'Word2Vec'.

**2.2.1 Literature review findings.** For the purpose of the literature review we evaluated each of the 52 articles to assess whether the authors had investigated biases within their model, specifically looking at word embeddings. We found that only two papers of 52 had assessed such biases within the NLP model architecture. The results of Section 3.0 are therefore concerning as we demonstrate clear biases in Word2Vec and GloVe embeddings which have been used in the research studies throughout the reviewed literature. The review of Clinical NLP by Kalyan et al. highlight the use of social media datasets, and models that are trained on existing word embeddings such as GloVe and Word2Vec [52]. Another example by Coppersmith et al. describes an automated model that estimates suicide risk from social media data, using pretrained GloVe embeddings for initial development [10]. We have also seen NLP workshop tasks that use of GloVe and Word2Vec for developing models to screen for suicide based on Reddit data [53]. GloVe and Word2Vec are two of the most widely used word embedding models and we therefore choose to analyse these embeddings specifically in our own primary analysis in Section 3.0.

## 2.3 Interpretation of results

The final stage of NLP development that we explored in our literature review is the 'Interpretation of results'. There exists a large body of research that describes the implicit biases of health providers and the means by which this affects patient care [15,21–22]. For example, existing mental health biases have resulted in men and women with equivalent levels of pathology being diagnosed at different rates [15]. In patients with the same presenting symptoms of psychiatric trauma, physicians are more likely to diagnose personality disorders in females while diagnosing PTSD in their male counterparts [22]. As a result, we see medical databases that suggest women have higher rates of personality disorders, when in truth practitioner bias has

resulted in a skewed representation of true organic disease [17–22]. To create effective digital mental health care we must consider the implication of bias in our historic medical datasets, and also consider the biases of the newer practitioners who are developing the NLP models of AI mental health care.

Firstly, as NLP models are based on datasets that have been affected by bias, these datasets must be critically appraised by specialists to determine whether the dataset reflects true organic disease. A good example recently has been provided by Vyas. et al, who provide a comprehensive review of clinical algorithms that encode biases as a result of the historically flawed datasets [54]. The recommendations of this review are essential for any digital health developer creating an NLP model who must first question whether their base data is influenced by historic biases.

Secondly, the developers of digital health models are susceptible to the same implicit biases that have led to our discrimnatory datasets. The implicit bias of a practitioner may impact the decisions they make based on the output of an NLP model e.g. whether to act on a suicide risk score or not. Medical guidelines often rely on clinical risks scores, however these are combined with subjective clinical judgement which may be influenced by unconcious bias [54]. The decisions made on the basis of NLP models must therefore be analysed to see if different demographic groups are treated equally. Lastly, the developers who are integral to the model design may introduce their own subconscious prejudices.

The literature review revealed a number of NLP models that relied on 'reviewers' to annotate datasets. These reviewers train the datasets by labelling data with different sentiment values. The biases of these individuals may affect their annotation of data, impacting the downstream model results. To build better models researchers must first challenge existing medical dogma and integrate the more recent understandings of biased medical practice in order to mitigate the propagation of these effects.

**2.3.1 Literature review findings.**  One study out of the 52 discussed the bias of practitioners, referring specifically to the potential biases of annotators involved in labelling the training data [55].

## 3.0 Primary analysis of word embeddings

In Section 2.2.1 we discussed how word embeddings such as GloVe and Word2Vec are used to create NLP models. Despite the widespread use of these embeddings, very few research papers have explored biases within these models. To expand on this we have performed a primary analysis of GloVe and Word2Vec embeddings, specifically relating to mental health.

Neural network language models represent words as high-dimensional vectors [56]. In these models individual words are converted to numerical vectors, and the relatoinships between words can be calculated as mathematical distances. 'Word embeddings' describe these representations of words as co-ordinates in vector spaces. As described in Section 2.2 these embeddings can be analysed to assess how words relate to one another. For example, Bolukbasi et al. demonstrated biases in vector spaces through the use of analogy questions. Bolukbasi et al. explored the vector relationships between gendered terms ('man'/'woman') and occupations, leading to the paper "*Man is to Computer Programmer as Woman is to Homemaker? Debiasing Word Embeddings*" [49]. Our method undertakes a similar analysis for exploring mental health biases in word embeddings.

We chose to examine 'GloVe' embeddings and 'Word2Vec' embeddings as these were most frequently cited in the literature and are often used to develop mental health models from online data [52]. In their systematic review of clinical uses of NLP, Kalyan et al. describe four categories of data used in clinical NLP models: Electronic health records,

Social Media Corpora, Online Medical Knowledge Sources and Scientific Literature [52]. The NLP models derived from social media corpora often use Word2Vec and GloVe. Alternate models also exist that build embeddings from medical databases and the scientific literature, however for this paper we focus on the use of Word2Vec and GloVe, as opposed to the narrower datasets described in more detail in the paper by Kalyan et al [52]. As described by Pennington et al. GloVe embeddings were trained on text copora from Wikipedia data, Gigaword and web data from Common Crawl which built a vocabulary of 400,000 frequent words [57]. Word2Vec was trained on the Google News dataset (containined ~ 100billion words) which resulted in a model of 300-dimensional vectors for 3 million words and phrases [58].

Research has demonstrated that the field of psychiatry exhibits significant biases which result in patient's receiving different diagnoses/treatment due to their demographic background [15–22]. Whether this is due to gender, race, sexuality or other factors, researchers have demonstrated the influence that identity plays on the care that a patient receives. For this reason we chose to explore relationships between demographic labels and psychiatric terms within the word embeddings. By illustrating the relationship between certain diagnoses/mental health features and different demographic terms, we aim to highlight any pre-existing biases within these vector spaces.

## 3.1 Method of primary analysis

Analogy completion via vector arithmetic is a popular method for exploring relationships within word embeddings [48]. This is the approach adopted by Bolukbasi et al. to explore the gender biases relating to occupation in their paper "*Man is to Computer Programmer as Woman is to Homemaker*? *Debiasing Word Embeddings*" [49]. The 'complete-the-analogy' question takes two existing words with a defined relationship and then uses this relationship to generate a fourth word from a third word [48–49]. This method therefore adopts an open-ended approach that allows us to pose an open question to the model (e.g. Man is computer programmer, as woman is to?) which returns a result (in this case 'homemaker' was returned). As our paper is an initial inquiry into mental health biases, we chose the anaology technique as this allowed us to pose open questions and elicit biases which may have remained undetected. Alternative methods do exist that provide a more rigorous quantative analysis of biases in word embeddings, such as the WEAT method which can also assess for statistical significance [59]. We selected the open-ended analogy approach over this as it allows for a wider scope of discovery. As a result, we cannot present findings of statistical significance, however to account for this we have repeated our analysis across different model dimensions to demostrate the stability of our findings (see Section 3.4).

## 3.2 Anaology methodology

Analogies are performed as a set of four words written as [48,49,56,60]:

$$w1 : w2 : w3 : w4$$

For these four words the relationship between word *w1* and *w2* is the same as the relationship between *w3* and *w4*. Therefore, when considering the vector coordinates of each word we can assume that [56,60]:

$$w2 - w1 = w4 - w3$$

Rearranged to:

$$w4 = w2 - w1 + w3$$

To create the 'complete-the-analogy' question in the word embeddings [48,56,60]:

1. We take the two words $w1w2$ that have a defined relationship between them.

2. We then select a third word $w3$ and request a fourth word $w4$, which is related to $w3$ by the relationship defined above.

A common example of this process is [49,60]:

$$King(w1) : Queen(w2) : Man(w3) : Woman(w4)$$

$$King\,is\,to\,Queen\,as\,Man\,is\,to\_\_?(w4)$$

$$Queen(w2) - King(w1) = w4 - Man(w3)$$

$$w4 = w2(Queen) - w1(King) + w3(Man)$$

$$w4 = Woman$$

Research has demonstrated that the use of these analogies to assess vector spaces produces a high accuracy for building lexical representations [48,49,56,60]. We use this approach to explore the relationships between psychiatric terminology and different demographic labels. Our first example in Section 3.2.1 takes the analogy question:

$$British\,is\,to\,Depression\,as\,Irish\,is\,to?(W4)$$

After running this function in Google CoLab the returned result was 'alcoholism'.

$$British(w2) - Depression(w1) = w4 - Irish(w3)$$

$$w4 = alcoholism$$

This example illustrates a diagnostic bias within the embedding which relates the term "Irish" to the diagnoses 'alcoholism'.

Our analogies of Word2Vec embeddings were carried out on Google CoLab using the pymagnitude package and vector similarity functions (see Section 3.1). In Section 3.3 we adapted our code to provide additional analysis that supports the findings of our vector analogy questions. Finally, in Section 3.4 we used Matplotlib's visualisation tool to illustrate these biases in a graphical form. In our assessment of bias we chose to examine a number of demographic characteristics including religion, nationality, race, gender, age and sexuality.

**3.2.1 Demographic labels.** The misuse use of race, sex, age and other features is criticized in biomedical research due to the inappropriate use of these labels, unequal representation of different social groups and the neglect of other key identity factors [61–63]. Despite this limitation, we have used these terms below as they reflect the current data collected by the UK & USA census bureaus and continue to pervade the medical field.

**3.2.2 Psychiatric terminology.** Mental health assessments are predominantly used for three purposes

1. To diagnose patients and for continued assessment.

2. To assess risk to self (self-harm/suicide).

3. To assess risk to others (violence/homicide).

We used these objectives as a guide for choosing which psychiatric terms to investigate. In the examples below we explore how diagnoses are related to different identity features and how specific characteristics are more closely related conceptualizations of risk.

### 3.3 Analysis of Word2Vec embeddings

The Word2Vec embeddings were used to assess for diagnostic bias using vector similarity functions on Google CoLab. The codebook detailing these functions are provided in the data repositry. We used similarity functions to extract the terms from the text corpus which were most closely related to the word we were interested in e.g. "depression". Each function used is highlighted at the top of each table. We have included the top three findings that was returned from each function.

In the second column of each table we have provided the numerical value of the vector similarity function for that term. The numerical values for each analogy are cosine distances that range from -1 to +1 and represent the vector distance between the two words examined e.g. "british" and "depression". A value of 1.0 is the point value of the original term (e.g. 'british), and hence if another term had a relative value of 1.0 this would be located at the same point in the vector space. How close a value is to 1.0 reflects how closely associated it is with the original term.

For each anaology question we combined the upper and lower case versions of an identifier term (i.e. "Teenager" and "teenager"). The full code for this can be found in the data repositry.

**3.3.1 Nationality and depression.** The analogy question "British is to Depression, as Irish is to _ (W4)?" returns the result 'alcoholism'. In the codebook in the data repository we have included additional analogy questions for other racial identifiers (e.g. American, Asian) if readers wish to explore these too.

**3.3.2 Race and depression.** The analogy question "White is to Depression, as Black is to _ (W4)?" returns the result "undergone_electroshock_therapy'. In the codebook in the data repository we have included additional analogy questions for other racial identifiers (e.g. Latinx) and invite the readers to look at these too.

**3.3.3 Gender and depression.** The analogy question "Man is to Depression, as Woman is to _ (W4)?" returns the result 'perinatal_depression'. In our supporting codebook we have included additional analogy questions for other gender identifiers (e.g. trans, girl, boy).

**3.3.4 Religion and depression.** The analogy question "Christian is to Depression, as Atheist is to _ (W4)?" returns the result 'hypochondria'. In the codebook in the data repository we have included additional analogy questions for other religious identifiers (e.g. Muslim, Jew, Sikh).

**3.3.5 Age and depression.** The analogy question "Grandparent is to Depression, as Adolescent is to _ (W4)?" returns the result 'Anxiety_Disorders'. In the codebook in the data repository we have included additional analogy questions for other age identifiers (e.g. Child, Parent).

### 3.4 Comparison of diagnoses by demographic labels

In this section we continue to examine the Word2Vec text corpus. To expand on the analogy questions performed above, we wrote code to elicit the most closely related demographic term to different diagnostic labels in the word embeddings. To do this we took a list of psychiatric conditions from the Diagnostic and Statistical Manual of Mental Disorders (DSM-5),

including ADHD, alcoholism, anxiety, bipolar disorder (the complete list of diagnoses included is available in the supporting documents). We then took a demographic term relating to a specific theme (e.g. Woman, for investigating gender), and sought out the closest psychiatric term to this. For example, when analysing the term "queer", the most closely relating psychiatric term was 'substance_abuse' (See Table 3.4.1). Tables 3.4.1–3.4.3 demonstrate the mental health terms that are found to be most closely related to different demographic labels of race, nationality, ethnicity, and gender. We chose demographic labels that are currently in use by the UK & USA Census Bureaus. The demographic labels below have been selected as examples for the manuscript, a more comprehensive review of different demographic labels can be found in the data repositry.

**3.4.1 Gender and diagnostic bias.**

**3.4.2 Race and diagnostic bias.**

**3.4.3 Nationality and ethnicity, and diagnostic bias.**

## 3.5 Analysis of Glove 50d embeddings [64]

Section 3.0–3.3 presented our analysis of Word2Vec embeddings. Tables 3.3.1 to 3.3.5 show the results of our analogy questions, which demonstrate biases along the lines of gender, race, nationality and age. Section 3.4 will now focus on GloVe embeddings. GloVe embeddings are another text corpus that are used widely for NLP applications including for mental health models. In the following section we have illustrated the biases that exist within GloVe embeddings using a graphical technique, as opposed to the vector similarity functions demonstrated above. GloVe embeddings are available across different dimensional models. In this section we have presented the results from our analysis of the 50 dimension model, however we have included further analysis on additional dimensions in S3 Appendix in the data repositry. To create the following graphs we used Matplotlib in Google Colab, for which the full codebook can be found in the supporting documents.

These graphs illustrate the relationship between specific terms and a concept such as gender. To do this, a pair of opposing words is used on the X axis of the graph (e.g. 'she' and 'he') and another pair of opposing words is used on the Y axis (e.g. 'safe' and 'dangerous'). New words are then introduced to the space to see how they relate to these four other terms/two concepts. Graph 1 illustrates our first example of this. We use the terms 'She' and "He' on the X axis to reflect gender, and as expected we observe that the female gendered terms (e.g. 'mother') closer to the 'She' pole. In addition, we place the terms 'Safe' and 'Dangerous' on the Y axis, to see how each gendered term relates to the concept of safety (see Section 3.4.1). The distance between words on the graph reflects the mathematical distance that is found between those vector points in the text corpus.

In the following section we create graphs based on these principles, applied to psychiatry. In Section 3.2.2 we explain that psychiatric assessments are used to assess for 'risk to self' and 'risk to others'. To analyse biases in these risk assessments we explore how these concepts of risk may relate to different demographic features, such as in Graph 1 where 'safe' and 'dangerous' were used as the opposing terms on the axis for analysing 'risk to others'. We use this graphical

**Table 3.3.1. British is to depression, as Irish is to 'alcoholism'.**

| Mental Health Condition ('Irish') | Vector Similarity Value |
|---|---|
| 'alcoholism' | 0.4766507 |
| 'Seasonal_Affective_Disorder' | 0.4354807 |
| 'mental_illness' | 0.4294042 |

**Table 3.3.2. White is to depression, as black is to 'undergone electroshock therapy'.**

| Mental Health Condition ('Black') | Vector Similarity Value |
|---|---|
| 'undergone_electroshock_therapy' | 0.52358043 |
| 'depressive_illnesses' | 0.51143694 |
| 'Compulsive_gambling' | 0.4988289 |

**Table 3.3.3. Man is to depression, as woman is to 'perinatal_depression'.**

| Mental Health Condition ('Woman') | Vector Similarity Value |
|---|---|
| 'perinatal_depression' | 0.53862274 |
| 'post_partum_depression' | 0.52797025 |
| 'postpartum_mood' | 0.52278244 |

**Table 3.3.4. Christian is to depression, as Atheist is to 'hypochondria'.**

| Mental Health Condition ('Atheist') | Vector Similarity Value |
|---|---|
| 'hypochondria' | 0.49891692 |
| 'clinically_depressed' | 0.48913783 |
| 'manic_depressive_disorder' | 0.48164117 |

**Table 3.3.5. Grandparent is to depression, as Adolescent is to 'Anxiety_Disorders'.**

| Mental Health Condition ('Adolescent') | Vector Similarity Value |
|---|---|
| 'Anxiety_Disorders' | 0.53548015308 |
| 'alcoholism' | 0.5293563738 |
| 'Addiction' | 0.50526595 |

space to map out how different demographic identifiers (e.g. 'mum', 'dad') relate to the opposing features of 'safe' and 'dangerous'. We then used synonyms to assess for the consistency of our findings, as in Graph 2 which uses 'violent' and 'innocent' instead of 'safe' and 'dangerous'.

**Multi-dimensional analysis.** To further assess for the stability and the significance of these results, we repeated our analysis on the 200d (200 dimension) and 300d (300 dimension) versions of the model. The trends observed in the 200d and 300d models reflect our findings below (e.g. the gendered trend of 'risk to others', which places female identifiers closer to the term 'safe'). S3 Appendix provides the graphs of this additional analysis and the full codebook

**Table 3.4.1. The closest mental health term to each gender label.**

| Gender Label | Most Closely Related Mental Health Diagnosis | Vector Similarity |
|---|---|---|
| Queer | substance_abuse | 0.20108144 |
| Gender_fluid | obsessive_compulsive | 0.11202571 |
| Man | bi_polar_disorder | 0.11507569 |
| Cis_gender | substance_abuse | 0.220669036 |
| Trans_gender | anxiety_disorders | 0.275229517 |
| Woman | alcholism | 0.098164104 |

**Table 3.4.2. The closest mental health term to each <u>racial</u> demographic label.**

| Race Label | Most Closely Related Mental Health Diagnosis | Vector Similarity |
|---|---|---|
| Latino | substance_abuse | 0.22431692 |
| African_american | schizoaffective_disorder | 0.1818381 |
| Native_american | substance_abuse | 0.2724196 |
| Asian | compulsive_hoarding | 0.0947723 |
| Hispanic | ADHD | 0.17809318 |
| White | alcoholism | 0.11180493 |
| Black | bipolar_disorder | 0.12816364 |

**Table 3.4.3. The closest mental health term to each <u>ethnicity</u> label.**

| Ethnicity Label | Most Closely Related Mental Health Diagnosis | Vector Similarity |
|---|---|---|
| Irish | alcoholism | 0.14398772 |
| African_american | schizoaffective_disorder | 0.1818381 |
| American | obsessive_compulsive | 0.07114809 |
| Chinese | obsessive_compulsive | 0.13045943 |
| Italian | obsessive_compulsive | 0.103272386 |
| Polish | alcoholism | 0.098870605 |
| German | obsessive_compulsive | 0.07460512 |
| English | schizoaffective_disorder | 0.12912744 |
| Asian | compulsive_hoarding | 0.0947723 |

of our multi-dimensional analysis can be found in the data repository. All graphs listed below are also available in the data repositry in higher resolution.

Lastly, it is important to note that in both graphs below our choice of vocabulary limits the patterns that we observe. As we have chosen to examine the American-English Language focusing on the 'Standard English' vernacular, this excludes the terminology of a wide range of cultures and population groups. As described in Section 2.1, our language is determined by our position in society and for this reason the vocabulary used in these graphs is limited.

**3.5.1 Gender bias—Risk to others.** Graph 1 is set up with the vectors of words 'safe' and 'dangerous' forming the opposing poles of the Y axis, and words 'he' and 'she' forming opposing poles of the X axis. Gendered family terms have then been plotted on the graph to demonstrate how these different labels vary in the relationship to the concepts of 'safe' and 'dangerous'. From this we can see that there is a gender bias which places the female terms closer to the word 'safe'.

**Graph 1–50 dimension GloVe Analysis of Gender Bias and Risk to Others (terms 'safe' and 'dangerous')** (*See* S3 Appendix).

**Graph 2–50 dimension GloVe Analysis. Gender Bias and Risk to Others (terms 'violent' and 'innocent').** Graph 2 is set up with the same concepts as Graph 2, however we have used the alternate adjectives 'violent' and 'innocent' on the Y axis. From these we can see that there is a gender bias which places the female terms closer to the word 'innocent'. Of note, in S3 Appendix we present the results of this analysis for the 200d and 300d models where the findings are not consistent. We see that the terms 'mum' and 'aunt' are positioned further away from the 'innocent' pole, and the terms 'brother', 'uncle' and 'son' are positioned closer to the 'innocnet' pole. This highlights the importance of investigating individual versions of a model, to elicit specific biases. (*See* S3 Appendix).

**3.5.2 Race—Risk to self.** Graph 3 illustrates vector relationships between different racial terms and the concept of suicide. Similarly to Table 3.4.2, we have used terms which most closely reflect those recorded by the UK and American Census Bureaus.

In this example we have given the poles of both the X and the Y axes the same terms of "Suicide" and "Healthy". This allows us to focus on how different racial terms relate to just the one concept of suicidality. As the axes use the same terms and units, the scales are the same. Therefore the central gradient line can be interpreted as a 2D line, which illustrates the points of each term relative to either pole. Graph 3 is available in the repositry at higher resolution, but for clarity the order of assocation from 'suicide' to 'healthy' progresses as follows (1) african_-american, (2) native_american, (3) asian_american, (4) black, (5) white, (6) latinx, (7) asian, (8) hispanic.

**Graph 3–50 dimension GloVe Analysis. Racial Bias and Risk to Self (terms 'suicide' and 'healthy')** (*See* S3 Appendix).

**3.5.3 Sexuality, gender and diagnostic bias.** Graph 4 illustrates diagnostic bias on the basis of sexuality and gender. On the X-axis we use the opposing poles "Heterosexual' and 'Homosexual' to look at sexuality, and on the Y-axis we use 'Cis_gender' and 'Trans_gener' to incorporate gender identity. We have then introduced new psychiatric diagnostic terms into the space (e.g. "PTSD"), to observe how each diagnoses relates to the four poles. Of note, we cannot establish whether these biases reflect existing population trends as both 'heterosexual' and homosexual' are highly heterogeneous populations and mental health risks vary widely amongst both groups. Futhermore, the terms we have used are limited and do not represent the scope of terms used to self-identify in the LGBTQ+ community. In attempt to compensate for this we used a wider range of terms in Table 3.4.1, however we acknowledge that this is not fully inclusive in its language.

The analysis presented in Graph 4 was repeated across the 200d and 300d dimensions of GloVe, for which the following findings were consistent:

1. The terms 'substance_use', 'paranoia' and 'suicide' were clustered towards the term 'Homosexual'.

2. The top left field is largely empty of diagnostic terms, which is the area corresponding to 'Cis_gender' and 'Heterosexual'.

3. The exception to Point 2, is that 'PTSD' is consistently focused towards 'Cis_gender' and 'heterosexual'.

**Graph 4–50 dimension GloVe Analysis. Sexuality, Gender and Diagnostic Bias (terms 'homosexual', 'heterosexual', 'trans_gender', 'cis_gender')** (*See* S3 Appendix).

## 4.0 Discussion

The presence of bias at any stage of model development risks creating tools that disadvantage certain patient groups. As a result data scientists need to widen their definition of success from focusing on model accuracy and result reproduction. The integration of medicine and data science must acknowledge the medical and social biases that underlie these models and work to unpack existing dogma before building new tools for the future.

## 4.1 Expression of data—Recommendations

Of the 52 articles reviewed in the literature, only two discussed the demographic variation of language expression within their cohort. Yet, as highlighted by Lakoff two speakers may be describing exactly the same thing, but their descriptions may end up sounding entirely

different [65]. Applied to mental health, this holds significant implications. A man and a woman may experience the exact same sentiment/intensity of feeling but the variation in their expression will limit who the NLP models detects.

**4.1.1 Integration of transcultural psychiatry.**   NLP models may entirely miss out cultures who express their suffering through vocabulary that differs from the existing standards found in the medical literature. For developers that adopt a lexical approach, dictionaries that pay particular attention to culturally different expressions of mental health will be beneficial. Kohrt et al. provide a systematic review of transcultural psychiatry and the different cultural idioms of distress that warrant greater attention [33]. Integrating such vocabulary into the dictionaries of lexical models could mitigate the negative impact of unequal cultural representation in existing psychiatric literature.

**4.1.2 Bias of describing clinicians.**   A number of the articles explored the use of NLP in electronic health records (EHRs) and medical notes in attempt to identify risk factors for suicide from the text data. The text used in these studies is therefore provided by the clinician, and bias refers to the written language of the provider. The use of unstructured clinical notes for NLP-based prediction tools is emerging as a useful tool for identifying certain health conditions. However, these tools also require an examination of the bias that occurs when clinicians describe the experiences of different patient groups. In models that use this form of language analysis it would be useful to explore associative relationships between clinician descriptions of patient's and the patient's demographics.

**4.1.3 Reliance on self-disclosures.**   As noted by Calvo et al, researchers utilize different means for identifying 'positive cases' i.e. those with diagnosed mental illness [3]. Some use direct self-reporting (disclosure on a technology platform), whereas others use indirect self-reported (via behavior, e.g. joining a support group) [3]. This immediately raises a question of validity. Are these studies phenotyping organic mental illness or are they simply phenotyping a subset who are more likely to self-disclose?

The work of Choudhury et al. that obtained clinically accurate data by crowdsourcing and inviting users to complete a mental health survey could provide a more evidence based method for obtaining this data [8]. Furthermore, Choudhury's work which follows up on the model by correlating results with a qualitative analysis of the surveyed patients demonstrates a positive approach to assessing the internal validity of a model.

**4.1.4 Domain specific datasets NLP research.**   Denecke et al. describe the importance of domain-specific datasets in the context of sentiment analysis [2]. Sentiment analysis is often domain dependent and consequently lexicons need to be adapted to domain-specific interpretations of words, this is especially important in psychiatry which is highly context dependant [2]. Wang et al demonstrate the superior performance of medical AI models that use embeddings trained on clinical notes and PubMed articles, as opposed to GloVe or Google News [47]. Such embeddings are better at capturing the semantic meaning of medical terms [47].

## 4.2 Analysis of data—Recommendations

The results of our primary analysis illustrate the importance of assessing bias in word embeddings. Other models that take lexical approaches may avoid the biases found in vector spaces, however these researchers will still face the challenges of differential expression and biased interpretation.

**4.2.1 Statistical fairness.**   Statistical fairness is a rapidly expanding research discipline. Machine learning scientists are at the front end of exploring methods to mitigate unfair models that impose different outcomes on different populations at different rates [66]. By exploring metrics such as false positives, false negatives and statistical polarity across different datasets,

developers can assess for fairness within their model [66]. Cheouldechova & Roth provide a comprehensive review of the methods being used to take both statistical fairness and individual fairness into account during model development, the extent of which are beyond the scope of this paper [66].

**4.2.2 NLP De-biasing techniques.**   Sun et al. offer a summary of de-biasing techniques for NLP models [67]. Within their review they discuss methods for both assessing models for bias (e.g. through variations of implicit association tests), and methods for mitigating biases [67]. Most of these tools can be tailored to mental health and used to explore demographic biases in psychiatric NLP tools. The integration of these methods within existing NLP models can mitigate potential damaging effects [67]. It should be noted that these debiasing methods suffer their own limitations as outlined by Gonen et al [68]. These authors describe how these techniques can be misleading. Employing such measures can appear successful but actually result in simply hiding biases that continue to persist in the dataset. Researchers must be cognisant of these limitations. Furthermore, to expand on the work of Sun et al further, more research is needed into mitigating word embedding bias that is specific to mental health.

## 4.3 Interpretation of results—Recommendations

Throughout the literature review little attention was given to the biases of professionals labelling or interpreting the model. This was particularly noted in the annotating of text corpora where only one paper discussed the implicit biases of the annotators. For accurate and representative datasets, these professionals must also be trained on their own biases and data must be produced by balanced demographic groups.

**4.3.1 Reflexivity statements.**   The integration of 'reflexivity statements' which are common in qualitative research would also help address this concern. Reflexivity statements allow researchers to provide a brief statement on how their own values and identity may impact the impartiality of their work. Through the analysis of ones own role, a researcher should self-critique and self-appraise how their own experience may influence stages of the research process [69]. Maura Dowling extends on this further in her paper "Approaches to reflexivity in qualitative research" [69].

**4.3.2 Cross-disciplinary teams.**   The literature review was carried out across six disciplinary databases between which there was only one crossover of research papers. The isolation of knowledge within disciplinary silos is a barrier to comprehensive research and model development. The most effective NLP models will benefit from the expertise of linguistics, data scientists and content experts who are trained in the underlying biases and history of their specialized domain.

## Conclusion

Cathy O'Neil states that the most dangerous algorithms "define their own reality and use it to justify the results" [70]. Medicine has been doing this for years without the application of AI. The interplay of historical biases, sample bias and knowledge-based biases, have resulted in the health disparities we continue to see today [15–28]. We take our assumptions of 'truth' based on a biased science, we then reinforce this 'truth' by engaging in biased practice and then we attribute our findings as endogenous to a patient group. If we do not pause to question the existing 'reality' of medicine, we will create AI that will project health disparities into the future. For applications of Natural Language Processing in mental health, this requires a rigorous assessment of the differences in our existing language expression, the biases that are present within NLP models and the unequal care of different patient groups in current clinical

practice. AI provides us with an opportunity to reflect on existing medical dogma, unpack it and build new models that will improve the future care of our patients.

## Supporting information

**S1 Appendix. MeSH terms and results of literature review.**
(DOCX)

**S2 Appendix. Articles included in literature review.**
(DOCX)

**S3 Appendix. The 50 dimension model is the original model used for creating graphs 1–4 in the manuscript.** Here we have repeated the analysis performed above with the 50d model to demonstrate consistency of trends.
(DOCX)

**S1 File.**
(PY)

**S2 File.**
(PY)

## Author Contributions

**Conceptualization:** Isabel Straw.

**Data curation:** Isabel Straw.

**Formal analysis:** Isabel Straw.

**Investigation:** Isabel Straw.

**Methodology:** Isabel Straw, Chris Callison-Burch.

**Project administration:** Isabel Straw.

**Supervision:** Chris Callison-Burch.

**Validation:** Chris Callison-Burch.

**Writing – original draft:** Isabel Straw.

**Writing – review & editing:** Isabel Straw, Chris Callison-Burch.

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
