## [Decision Letter · Decision Letter 0]

20 Mar 2020

PONE-D-20-06032

American is to depression as Irish is to alcoholism: Artificial Intelligence in mental health and the biases of language based models.

PLOS ONE

Dear Dr Straw,

Thank you for submitting your manuscript to PLOS ONE. After careful consideration, we feel that it has merit but does not fully meet PLOS ONE’s publication criteria as it currently stands. Therefore, we invite you to submit a revised version of the manuscript that addresses the points raised during the review process.

The reviewers are in favor of asking the authors for a revised manuscript with technical and communication upgrades. Please address their comments in whole.

We would appreciate receiving your revised manuscript by May 03 2020 11:59PM. To enhance the reproducibility of your results, we recommend that if applicable you deposit your laboratory protocols in protocols.io, where a protocol can be assigned its own identifier (DOI) such that it can be cited independently in the future. For instructions see: http://journals.plos.org/plosone/s/submission-guidelines#loc-laboratory-protocols

We look forward to receiving your revised manuscript.

Kind regards,

Christopher M. Danforth

Academic Editor

PLOS ONE

Journal Requirements:

2. Please include in your appendix the date when the systematic literature search was carried out.

3. Thank you for stating the following in the Competing Interests section:" I have read the journal's policy and the authors of this manuscript have the following competing interests:

Isabel Straw has been an unpaid intern with the e-Health company "Neuroflow" as part of her Masters in Public Health (MPH) Program. While this has been entirely separate from this research project, the timing of the internship overlaps with the period of time taken to write this article.

Chris Callison-Burch declares no competing interests."

Reviewers' comments:

Reviewer's Responses to Questions

**Comments to the Author**

1. Is the manuscript technically sound, and do the data support the conclusions?

Reviewer #1: Partly

Reviewer #2: Yes

2. Has the statistical analysis been performed appropriately and rigorously? 

Reviewer #1: No

Reviewer #2: No

3. Have the authors made all data underlying the findings in their manuscript fully available?

Reviewer #1: Yes

Reviewer #2: No

4. Is the manuscript presented in an intelligible fashion and written in standard English?

Reviewer #1: Yes

Reviewer #2: Yes

5. Review Comments to the Author

Reviewer #1: I like the paper and the extent to which it describes how and where the biases may appear in various stage of a machine learning model. It is important to keep in mind that the model would learn from the data it is fed on and it is our responsibility to make sure the “reality" that the data represents does not favor some groups more than others! The paper does a good job getting that point across.

That being said, there are few points I would like to highlight:

- Authors do not go into details of word embeddings they used for the analogy task and similarity tasks.

- The dimensions of word embeddings play an important role. It would be helpful to see more details and comparison across various dimensions of word embeddings

- Under the analogy tasks, authors do not take into account the capitalization of words which may change the results

- For example on line 317: the analogy shown is "American is to Depression as Irish is to Alcoholism" while in the equation in table under the line 317: vectors are for lower cases words!

- Section 3.2 presents some interesting results about the individual risk’s to themselves and their risk to others. The graphs capture the biases with respect to religion, race, gender, nationality but the details to capture the biases were not clear. It would have helpful to have more details so that the results are reproducible.

- Finally, It would be interesting to see if the biases hold in embeddings trained on domain specific data such as clinical notes and PubMed articles as mentioned in section 4.1.4

Overall, the paper presents an important case about highlighting the biases in language models and there is a need of collaboration among data scientists and medical experts to come together to build models that do not encode and further propagate the biased reality that data represents today!

Reviewer #2: This article provides an overview of applications of word embeddings to psychology studies. It highlights the biases that can be involved in such methods and cautions practitioners to consider specific points in the scientific process at which bias can be introduced. This is an important message for both the NLP and the psychology research communities, and I think that this literature review is of great value.

The article would be greatly improved and its message would be much stronger if some technical concerns were first addressed. With these revisions, I believe this would be a very strong contribution.

My major concerns are the following:

1. What training data was used for the word2vec and GloVe models? This needs to be specified, particularly since one of the core arguments of the paper is around biases in the training data. I’m guessing that these are large, pre-trained models (but I don’t know for sure), in which case, does this actually mimic the setting in which you expect word embeddings to be used for psychology studies? In other words, wouldn’t you expect these studies to train their own models on their particular datasets of interest (perhaps social media, as mentioned earlier in the paper)? Citing Google Colab isn’t enough, since I can’t reproduce your work with only this knowledge.

2. What significance tests were done for the results in Tables 3.1.1-3.1.5 and in Graphs 1-4? How stable are these results? If you chose not to perform significance tests (perhaps because the model is very large and pre-trained), this should be explained.

3. I’m having trouble understanding Graphs 1-4. Why are the x and y axes showing the same measurements? What is the purpose in representing the results in this way? The diagonal lines don’t mean anything, so perhaps a 2D line or a barplot with error bars would be a better representation.

4. For Tables 3.1.1-3.1.5, why did you choose to use analogies rather than showing the most similar words for each demographic group or using a test like WEAT (Caliskan et al., 2016)? How were the seed term pairings decided (e.g. why is “depression” matched with “american” rather than “irish”?), and were the seed terms allowed to be returned in the results?

5. What papers were used in this literature review? The query terms are provided, but the paper titles are not. It would be helpful in understanding the results of the literature review.

6. The scope of the paper should be clarified in a few places to emphasize that this paper focuses specifically on word embedding models and not on other types of NLP techniques.

Some minor suggested changes:

Line 87: “human emotions” change to “linguistic representations of human emotions”

Line 128: I think there should be another paragraph here about how the medical data previously described is used to train machine learning models, and so these models might also be biased. This is the central motivation of the paper, so I think it should be explained clearly and strongly in the introduction.

Line 136: Reddit, Facebook, and Twitter aren’t the usual training sets for the embeddings used in the usual, big, pre-trained models. They are often used in computational social science research.

Line 144: This information is in the Appendix, but I think it’s worth briefly mentioning in the text at least which venues the data was pulled from (e.g. ArXiv, …)

Line 183-5: I think you could spell this out more clearly: "Models trained on data written mostly by white men will not accurately represent people in other demographic groups."

Line 197: The distributional hypothesis is used in many models outside of deep learning. It’s not a contribution of or unique to deep learning. Nor is the concept of finding distances between word vectors a contribution of or unique to deep learning.

Line 208: Does that research emphasize the model or the training data? There is some work showing that such biases can actually be magnified by the model, but my summary would be that most work focuses on effects of the biased training data (the bias is introduced through the data, and the model simply encodes that bias).

Section 2.2: I think this section could be extended and included more references. It’s not totally clear what “data interpretation” actually means. Later you mention both understanding model results and labeling training data as “data interpretation” and I need more explanation to buy the combination of these separate tasks.

Line 343: These two graphs are supposed to validate each other, but the words significantly change in order. For example, in the second graph, “sister” is now much more similar to “brother” “father” “uncle” than “mum” or “aunt”.

Line 368: What is data “expression”? I think this section and its corollary in the literature review should be labeled “training data” or something clearer.

Line 409-10: Embeddings trained on PubMed and clinical notes are indeed better at representing medical terms, but they are not necessarily better at representing the conversational and social data that has been cited in the paper and a prime use case for NLP models for psychology.

Line 410: I would add “use training data produced by balanced demographic groups” to the recommendations in this section (or if you disagree, explain). Domain specific data addresses part of this concern, but even within domain, we also want our data to be produced by balanced demographic groups.

Section 4.2.2: This section should also call out the extensive weaknesses of these “debiasing” methods. See, for example: Gonen, Hila, and Yoav Goldberg. 2019. “Lipstick on a Pig: Debiasing Methods Cover up Systematic Gender Biases in Word Embeddings But Do Not Remove Them.” arXiv [cs.CL]. arXiv. http://arxiv.org/abs/1903.03862.

Section 4.3.1: What are reflexivity statements? Some more detail here would be useful.

6. PLOS authors have the option to publish the peer review history of their article (what does this mean?). If published, this will include your full peer review and any attached files.

Reviewer #1: Yes: Sandhya Gopchandani

Reviewer #2: No

---

## [Author Response · Author response to Decision Letter 0]

10 Aug 2020

Dear Reviewers,

Thank you for your comments and recommendations for our manuscript “British is to depression as Irish is to alcoholism: Artificial Intelligence in mental health and the biases of language based models”. You will note that this title has changed from the title of the first draft, with ‘British’ replacing ‘American’. This resulted from our incorporation of the reviewer’s suggestions into the results section, and the consequent changes that are outlined below.

In our letter below we have responded to all of the points and recommendations from both reviewers, giving reference to the relevant line numbers of the manuscript where necessary. With submission of this revision we have also added the requested data and supporting documents to the data repository, including:

1. Literature review with details of each returned paper.

2. Codebook1: Our codebook demonstrating the code and the outputs for the results presented in Section 3.1 and 3.2.

3. Codebook 2: Our codebook demonstrating the code and the outputs for the results presented in Section 3.3.

4. Graphs: High-resolution images of all graphs included in the manuscript.

Thank you for your time and for your contribution,

Sincerely,

Isabel Straw and Chris Callison-Burch 

1.1 Reviewer 1 Overall Comments and Recommendations:

1.1.1 Authors do not go into details of word embeddings they used for the analogy task and similarity tasks.

Lines 320 – 329 have been added to give details on the word embeddings that we have used. We outline the training data used for these embeddings and reference the relevant studies/texts. 

1.1.2 The dimensions of word embeddings play an important role. It would be helpful to see more details and comparison across various dimensions of word embeddings.

The second draft of our manuscript includes a revised results section, which incorporates an analysis across various dimensions of the word embeddings. Lines 501 – 508 specifically describe our additional analysis of 200 Dimension and 300 Dimension GloVe, the results of which are provided in Appendix 3. We have also added our full codebooks to the data repository, which allows readers to access our full analysis. 

1.1.3 Under the analogy tasks, authors do not take into account the capitalization of words which may change the results

We have introduced a new results section to this second draft of our manuscript, which addresses this point. We rewrote our code to incorporate both upper and lower case versions of the words and have presented these results. As a result of our adjusted analysis, some of the results changed and so the examples we chose to present in the manuscript have also change. This includes the title, where we have replaced “American” with British”. Of note, all of the original analysis is still contained in the codebooks, which are available in the data repository. 

Reviewer 1 Specific Changes:

1.1.4 For example on line 317: the analogy shown is "American is to Depression as Irish is to Alcoholism" while in the equation in table under the line 317: vectors are for lower cases words!

Our results section now reflects our rewritten code and new analysis. All results provided are for both the upper and lower case versions of the words (see Point 1.1.3). 

1.1.5 Section 3.2 presents some interesting results about the individual risk’s to themselves and their risk to others. The graphs capture the biases with respect to religion, race, gender, nationality but the details to capture the biases were not clear. It would have helpful to have more details so that the results are reproducible.

With the changes to the results section we hope that these graphs are now clearer. The graphs in the first draft used an application on GitHub for analysing biases, however this application was not available for different dimensions of GloVe. Therefore we wrote our own code using Matplotlib, to illustrate these biases. We have provided a codebook in the data repository to allow the reproducibility of our results called “IStrawMultidimensionalGloVe”, which gives the full details of our code.

In response to a number of comments about the graphs now being clear, we have changed the axes on Graphs 1 – 4. As opposed to keeping both axes the same and essentially having a 2D plot, we have integrated four different terms across the two axes. For example, Graph 1 presents “She” & “He” on the X axis, and “Safe” and “Dangerous” on the Y Axis. This allows use to view new terms such as ‘mother’, in relation to all four of these words.

1.1.6 It would be interesting to see if the biases hold in embedding’s trained on domain specific data such as clinical notes and PubMed articles as mentioned in section 4.1.4

In Section 4.1.4 we provide an overview of the use of domain specific data. Unfortunately there is not scope in this paper to also include this area of research however this is fully referenced in the text. Lines 252 to 259 provide an overview of different datasets for Clinical NLP, Lines 319 to 329 then expand on this further. We also describe the importance of these types of datasets in our discussing in Lines 621 to 624.

2.1 Reviewer 2 Overall Comments and Recommendations:

The article would be greatly improved and its message would be much stronger if some technical concerns were first addressed. With these revisions, I believe this would be a very strong contribution.

2.2 Reviewer 2 Major Revisions

2.2.1 Training Data: What training data was used for the word2vec and GloVe models? This needs to be specified, particularly since one of the core arguments of the paper is around biases in the training data. 

Lines 321 – 329 have been added to give additional background on the Word2Vec and GloVe models. We have described the training data used for these models and referenced the key relevant texts. 

2.2.3 Does this actually mimic the setting in which you expect word embeddings to be used for psychology studies? In other words, wouldn’t you expect these studies to train their own models on their particular datasets of interest (perhaps social media, as mentioned earlier in the paper)?

To address this point we have included more information on the use of GloVe and Word2Vec in psychiatric NLP models throughout the manuscript. Specifically, Lines 252 – 259 have been added to describe relevant models that are trained on GloVe and Word2Vec. We have also referenced a paper by Kaylan et al. who provide a thorough review of data sources used for clinical NLP, including online data and pre-trained embeddings. In addition, we have highlighted further examples of models that are trained specifically on GloVe embeddings. 

2.2.4 Significance Test: I’m having trouble understanding Graphs 1-4. Why are the x and y axes showing the same measurements? What is the purpose in representing the results in this way? The diagonal lines don’t mean anything, so perhaps a 2D line or a barplot with error bars would be a better representation.

We have addressed these points in our new results section, which we have introduced to meet the technical concerns. Regarding the specific points above:

(i) Significance testing: Lines 348 to 356 have been introduced to justify our choice of method. In this section we explain that as we have chosen to perform analogy questions and similarity functions, we cannot perform the significance testing of more rigorous quantitative analyses such as WEAT. We chose to use analogy questions for this paper, as this allowed us to adopt an open-ended approach by posing a question and not setting the parameters. As an initial paper researching these biases, this allowed for a wider scope of recovery. In the future we hope to expand on this with more statistical analysis that builds on the results of our analogy questions.

(ii) Graphs 1 – 4: We have to change Graphs 1 – 4 of the manuscript, following our incorporation of a multi-dimension analysis of GloVe. In our first draft the graphs were created using an application on GitHub, however this was only available for the 50 dimension version of GloVe. Therefore to account for this, we wrote our own code with Matplotlib and created our own visualisation for the 50, 200 and 300 dimension versions of GloVe. 

In response to a number of comments about the graphs now being clear, we have changed the axes on Graphs 1 – 4. As opposed to keeping both axes the same and essentially having a 2D plot, we have integrated four different terms across the two axes. For example, Graph 1 presents “She” & “He” on the X axis, and “Safe” and “Dangerous” on the Y Axis. This allows use to view new terms such as ‘mother’, in relation to all four of these words. The only exception to this is Graph 3 which looks at racial bias, where we have kept the original graph as we felt this provided a good representation of different racial terms to the words ‘suicide’/’healthy’. We have added lines 543 to 545 to explain what the central line represents and how it can be interpreted.

2.2.5 Significance Test: What significance tests were done for the results in Tables 3.1.1-3.1.5 and in Graphs 1-4? How stable are these results? If you chose not to perform significance tests (perhaps because the model is very large and pre-trained), this should be explained.

(i) Significance Tests: As detailed in our response to point 2.2.4 we have added lines 348 to 356 to justify our lack of significance testing. Significance tests have not been performed, as our results are not statistical measures but descriptive results that illustrate the numerical distances between words in the vector space. We choose not to use the more advanced statistical techniques such as WEAT as we decided to employ the more open-ended approach of analogy questions (see point 2.2.4).

(ii) Stability of Results: Lines 353 to 356 have been added to describe how we use our multi-dimensional analysis to demonstrate stability of results. While we cannot perform significance testing we have taken these additional measures to demonstrate the stability of our findings: Upper/lower case versions of words, synonyms, multiple dimensions and the new analysis of Section 3.3 to support Section 3.2. 

2.2.6 For Tables 3.1.1-3.1.5, why did you choose to use analogies rather than showing the most similar words for each demographic group or using a test like WEAT (Caliskan et al., 2016)? How were the seed term pairings decided (e.g. why is “depression” matched with “american” rather than “irish”?), and were the seed terms allowed to be returned in the results?

(i) Choice of Method: Lines 348 to 356 have been introduced to justify our choice of method. In this section we explain that as we have chosen to perform analogy questions and similarity functions, we cannot perform the significance testing of more rigorous quantitative analyses such as WEAT. We chose to use analogy questions for this paper, as this allowed us to adopt an open-ended approach by posing a question and not setting the parameters. As an initial paper researching these biases, this allowed for a wider scope of recovery. In the future we hope to expand on this with more statistical analysis that builds on the results of our analogy questions.

In response to the second point, we have no incorporated the suggestion of using “the most similar words for each demographic group”. We have introduced Section 3.3 of the manuscript to cover this additional analysis, which supports Section 3.2.

2.2.7 What papers were used in this literature review? The query terms are provided, but the paper titles are not. It would be helpful in understanding the results of the literature review.

A full list of the literature review papers has been added to the manuscript in Appendix 2. Line 154-155 directs the author to this resource. Of note, there is a change to the manuscript here as the original manuscript had a row missing from the literature search table. We used the ACL Anthology database but had not separated these articles from the ArXiv papers. This has been corrected in the revised manuscript as detailed in lines 157 – 159 and Appendix 1.

2.2.6 The scope of the paper should be clarified in a few places to emphasize that this paper focuses specifically on word embedding models and not on other types of NLP techniques.

Lines 211 – 217 have been added to address this point. We briefly outline the history of NLP models and specify that we will be investigating word embeddings.

2.3 Reviewer 2 Minor Revisions

2.3.1 Line 87: “human emotions” change to “linguistic representations of human emotions”

This change has been made in the manuscript and can be found at Lines 85 – 86. 

2.3.2 Line 128: I think there should be another paragraph here about how the medical data previously described is used to train machine learning models, and so these models might also be biased. This is the central motivation of the paper, so I think it should be explained clearly and strongly in the introduction.

A new paragraph has been added to address this point. Lines 274 – 280 have been added to describe the history of biased datasets in medicine and the impact that his has on digital models. The discovery of biases in large medical databases is a new area of research and we have added Reference 63, which was published in June 2020 to provide the most recent literature on this.

2.3.3 Line 136: Reddit, Facebook, and Twitter aren’t the usual training sets for the embeddings used in the usual, big, pre-trained models. They are often used in computational social science research.

In Section 2.0 and 3.0 we have described and referenced a paper that describes the range of data sources that are used in clinical NLP. We outline the range of data that includes social media data, wider web data and medical data. We focus on Word2Vec/GloVe embeddings in our paper, however we also reference the other domain specific datasets that are described in more detail in Kalyan’s paper (Lines 319 – 329).

2.3.4 Line 144: This information is in the Appendix, but I think it’s worth briefly mentioning in the text at least which venues the data was pulled from (e.g. ArXiv)

Lines 157 – 159 describing the databases has been added to address this point.

2.3.5 Line 183-5: I think you could spell this out more clearly: "Models trained on data written mostly by white men will not accurately represent people in other demographic groups."

Lines 193 – 194 have been added to give more detail here, incorporating the point made above.

2.3.6 Line 197: The distributional hypothesis is used in many models outside of deep learning. It’s not a contribution of or unique to deep learning. Nor is the concept of finding distances between word vectors a contribution of or unique to deep learning.

Line 221 to 225 clarify this point and describe the use of the distributional hypothesis in NLP models. 

2.3.7 Line 208: Does that research emphasize the model or the training data? There is some work showing that such biases can actually be magnified by the model, but my summary would be that most work focuses on effects of the biased training data (the bias is introduced through the data, and the model simply encodes that bias).

We expand on this in Lines 234 to 242. This research focuses on biases that have resulted from encoded bias within the embeddings, such as in gender and ethic stereotypes. 

2.3.8 Section 2.2: I think this section could be extended and included more references. It’s not totally clear what “data interpretation” actually means. Later you mention both understanding model results and labelling training data as “data interpretation” and I need more explanation to buy the combination of these separate tasks.

To make this section clearer we have renamed the three stages of model development as: (1) Expression of Data, (2) Analysis of Data, and (3) Interpretation of Data. In Section 2.2 we have added Lines 140 - 150 to provide a clearer explanation of what we mean by the “Analysis of Data”. In this section we are referring specifically to how word embeddings may introduce bias into an NLP model. We give a brief overview of the evolution of word embeddings in recent years, to demonstrate the relevance of investigating these tools. 

2.3.9 Line 313: These two graphs are supposed to validate each other, but the words significantly change in order. For example, in the second graph, “sister” is now much more similar to “brother” “father” “uncle” than “mum” or “aunt”.

We have now changed the graphs in Section 3.4, such that this point does not relate to the new graph. The new versions of Graphs 1 – 4 were introduced in order to incorporate the multidimensional analysis, as explained in point 2.2.4.

The new versions of Graph 1 and Graph 2 support the same point made in our first draft where we illustrate that the female terms cluster towards the words ‘Safe’ and ‘Innocent’. While there is some rearranging over the terms, the overall trend holds.

2.3.10 Line 345: What is data “expression”? I think this section and its corollary in the literature review should be labelled “training data” or something clearer.

To make this section clearer we have renamed the three stages of model development as: (1) Expression of Data, (2) Analysis of Data, and (3) Interpretation of Data. Lines 168 - 179 provide a clearer explanation of what we mean by the “Expression of Data”. 

2.3.11 Line 409-10: Embeddings trained on PubMed and clinical notes are indeed better at representing medical terms, but they are not necessarily better at representing the conversational and social data that has been cited in the paper and a prime use case for NLP models for psychology.

In Lines 618 to 624 we have highlighted that embeddings trained on these datasets are most beneficial for understanding medical terms specifically. 

2.3.12 Line 410: I would add, “use training data produced by balanced demographic groups” to the recommendations in this section (or if you disagree, explain). Domain specific data addresses part of this concern, but even within domain, we also want our data to be produced by balanced demographic groups.

We have added lines 653 - 655 to the manuscript to address this point. 

2.3.12 Section 4.2.2: This section should also call out the extensive weaknesses of these “debiasing” methods. See, for example: Gonen, Hila, and Yoav Goldberg. 2019. “Lipstick on a Pig: Debiasing Methods Cover up Systematic Gender Biases in Word Embeddings But Do Not Remove Them.” arXiv [cs.CL]. arXiv. 

We have added Lines 643 – 649 to the discussion to include this point. We describe the limitations of debiasing methods and the impact this can have. We briefly outline how this can lead to inaccurate results and we signpost the author to the relevant text.

2.3.12 Section 4.3.1: What are reflexivity statements? Some more detail here would be useful.

We have added to lines 656 – 662 to explain this point better. We briefly describe reflexivity statements and provide the reader with a key reference text for learning further about this resource.

---

## [Decision Letter · Decision Letter 1]

25 Sep 2020

British is to depression as Irish is to alcoholism: Artificial Intelligence in mental health and the biases of language based models.

PONE-D-20-06032R1

Dear Dr. Straw,

We’re pleased to inform you that your manuscript has been judged scientifically suitable for publication and will be formally accepted for publication once it meets all outstanding technical requirements.

From an editorial point-of-view, we request that you change the manuscript's title to 'Artificial Intelligence in mental health and the biases of language-based models' i.e. to remove 'British is to depression as Irish is to alcoholism' as we feel this phrase could be perceived to be pejorative. Please make this change in your manuscript file and in the submission system alongside the other technical amendments.

Kind regards,

Christopher M. Danforth

Academic Editor

PLOS ONE

Additional Editor Comments (optional):

Reviewers' comments:

Reviewer's Responses to Questions

**Comments to the Author**

1. If the authors have adequately addressed your comments raised in a previous round of review and you feel that this manuscript is now acceptable for publication, you may indicate that here to bypass the “Comments to the Author” section, enter your conflict of interest statement in the “Confidential to Editor” section, and submit your "Accept" recommendation.

Reviewer #2: All comments have been addressed

2. Is the manuscript technically sound, and do the data support the conclusions?

Reviewer #2: Yes

3. Has the statistical analysis been performed appropriately and rigorously? 

Reviewer #2: Yes

4. Have the authors made all data underlying the findings in their manuscript fully available?

Reviewer #2: Yes

5. Is the manuscript presented in an intelligible fashion and written in standard English?

Reviewer #2: Yes

6. Review Comments to the Author

Reviewer #2: The authors have satisfactorily addressed all of my comments except significance testing. I'm still concerned about the lack of significance tests / error bounds, especially given the types of plots included it the draft. Claiming a linguistic association between a gender or sexuality and a mental health disorder should include some sense of the stability of these results, and since this paper is introducing these concepts to the mental health field, it's important to emphasize such tests and educate about their use. I disagree that it's not possible for the authors to perform some kind of stability test; for example, they could re-calculate the vector distances across permutations of the training dataset (maybe re-train the model on some medically relevant dataset. Or the authors could have used one of the many bias tests, like WEAT, that include permutation tests of the seed terms. Indeed, other papers also present untested results in the fashion presented in the paper, but that doesn't mean they were correct in doing so.

7. PLOS authors have the option to publish the peer review history of their article (what does this mean?). If published, this will include your full peer review and any attached files.

Reviewer #2: No

---

## [Editor Report · Acceptance letter]

9 Oct 2020

PONE-D-20-06032R1 

Artificial Intelligence in mental health and the biases of language based models. 

Dear Dr. Straw:

I'm pleased to inform you that your manuscript has been deemed suitable for publication in PLOS ONE. Congratulations! Your manuscript is now with our production department. 

Kind regards, 

on behalf of

Dr. Christopher M. Danforth 

Academic Editor

PLOS ONE